# Phototransistors Based on hBN-Encapsulated NiPS$_3$

**Yingjia Liu** [1,2] **and Xingdan Sun** [1,2,*]

1.  Shenyang National Laboratory for Materials Science, Institute of Metal Research, Chinese Academy of Sciences, Shenyang 110016, China
2.  School of Material Science and Engineering, University of Science and Technology of China, Hefei 230026, China
*   Correspondence: xdsun15s@imr.ac.cn; Tel.: +86-24-2397-1941

**Abstract:** Transition metal phosphorous trichalcogenides (MPX$_3$) have been extensively investigated as photodetectors due to their wide-bandgap semiconductor properties. However, the research involved in the photoresponses at low temperatures remain blank. Here, hexagonal boron nitride (hBN)-encapsulated NiPS$_3$ field effect transistors were fabricated by using the dry-transfer technique, indicating strong stability under atmospheric environments. The NiPS$_3$ devices with the thickness of 10.4 nm, showed broad photoresponses from near-infrared to ultraviolet radiation at the liquid nitrogen temperature, and the minimum of rise time can reach 30 ms under the wavelength of 405 nm. The mechanism of temperature-dependent photoresponses can be deduced by competition between Schottky barrier height and thermal fluctuation. Our findings provide insights into superior phototransistors in few-layered NiPS$_3$ for ultrasensitive light detection.

**Keywords:** two-dimensional materials; NiPS$_3$; vdWs heterojunction; phototransistor; Schottky barrier

## 1. Introduction

Phototransistors based on wide-bandgap semiconductors have exhibited favorable potential for applications such as military early warning, optical communication, remote sensing, and biochemical detection [1–5]. To date, conventional wide-bandgap semiconductors, specifically Ga$_2$O$_3$ [6–9], have always been considered as prospective candidates for practical commercial manufactures. However, it is still a challenge to improve the external quantum efficiency (EQE) owing to trapping of photogenerated carriers and poor absorption as their thickness reduced to atomic scale. In principle, the trap effect can be suppressed by passivating the surface dangling bonds of semicondutors [10], improving charge carrier mobility [11] and reducing the carrier transport distance [12].

Unexpected turnaround comes with the research boom of graphene [13], the majority of two-dimensional (2D) van der Waals (vdWs) materials, including graphene, tellurene, black phosphorus (BP), transition metal dichalcogenides (TMDs, bandgap range from 0.3 eV to 2 eV), which have been confirmed as having obvious photoelectric effects [14–16]. Compared to conventional phototransistors, new-generation photodetectors based on 2D materials demonstrate competitive advantages in overcoming the above obstacles for future applications. With regard to 2D materials, they display dangling bond-free surfaces, atomically smooth surfaces, atomically thin thickness and flexible assemblability, which implies their promising capability for achieving a state-of-the-art phototransistor possessed with high EQE, high sensitivity, and high responsivity. Among the extensive range of 2D materials are transition metal phosphorous trichalcogenides (MPX$_3$, M = Mn, Fe, Ni, Zn, and Cd; X = S and Se). Antiferromagnetic semicondutors [17] have been a new research focus recently because of their sufficiently cleavage energy [18] and broad tunable bandgap (1.2–3.5 eV) [19]. In particular, the emergence of this new 2D material family broadens the photo detection window to ultraviolet (UV) spectral at the 2D level, and their bandgaps can be tuned by changing transition metal and thickness. During the last decade,

several works have focused on the photoresponse of few-layered $MPX_3$ flakes. $FePS_3$ and $MnPSe_3$ are representative research objects, but studies concerning $NiPS_3$ are rare. The bandgap of monolayer and bulk $NiPS_3$ are calculated to be 3.01 eV [20] and 1.6 eV [21]. So far, He et al. investigated a UV photodetector based on $NiPS_3$ nanosheets (via chemical vapor deposition method), which shows an ultrafast rise time shorter than 5 ms together with a remarkable detectivity [22]. Zhang et al. revealed the great potential of $NiPS_3$ (prepared by electrochemical cathodic exfoliation) as a self-powered photo-eletrochemical detector [23]. Moreover, in addition to the research about linear polarization-resolved photoluminescence [24], there is no report about the photoresponsive behaviors at low temperature, and photocurrents in low temperature have a key role in studies of the Schottky barrier [25,26], which indicates a better understanding of physical phenomena at the interface between $NiPS_3$ and contact electrodes.

In this work, we comprehensively investigated the photoelectric performances of hBN-encapsulated $NiPS_3$ at different temperatures. The broad photoresponse from near-infrared to ultraviolet radiation was observed, and the rise time can reach 30 ms under the wavelength of 405 nm. To clearly rationalize the temperature-dependences in photoresponses, the carrier mobility, Schottky barrier height, and series resistance were also studied.

## 2. Materials and Methods

### 2.1. Materials

The high-quality bulk $NiPS_3$ single crystals were purchased from PrMat (Chemical vapour transport, 99.9998%, Shanghai, China). Few-layered $NiPS_3$ and hBN (crystals from HQ Graphene, Groningen, The Netherlands, about 15–25 nm thick) were exfoliated by using 3M (Saint Paul, MN, USA) Scotch tape [27]. $NiPS_3$ and hBN flakes were placed on a 1-mm-thick PDMS (polydimethylsiloxane) substrate and $Si/SiO_2$ substrates, respectively.

### 2.2. Methods

The hBN-encapsulated $NiPS_3$ heterostructures were fabricated in ambient conditions by using the standard dry-transfer method [28] with a homemade van der Waals transfer station. The PDMS was used as the medium to transfer the $NiPS_3$ flakes onto the target bottom hBN flakes, and the PDMS/polypropylene carbonate (PPC) stamp was used to transfer and release top hBN flakes to protect few-layered $NiPS_3$ from degradation.

A polymethyl methacrylate (PMMA) layer (495k MW, A4, MicroChem, Cape Town, South Africa) was spin-coated at 2000 rpm/min on the silicon substrate and baked at 180 °C for 7 min, another PMMA layer (950k MW, A2, MicroChem) was then spin-coated at 4000 rpm/min and baked at 180 °C for 2 min. The geometry of source/drain electrodes was defined by electron-beam lithography (EBL, Raith, Dortmund, Germany) and developing processes, and then the hBN under electrodes pattern was removed via reactive ion etching (RIE, Samco, Kyoto, Japan, $CHF_3$ and $O_2$ with a flow rate of 20 standard cubic centimeters per second (sccm) and 4 sccm respectively). Subsequently, Ti/Au (5/35 nm thick) contact electrodes deposited by a traditional thermal evaporation approach. The lift-off process was performed in two steps, first in acetone, and secondly in isopropyl alcohol. The electrical connections of final devices were in face contact.

### 2.3. Characterization

The bulk $NiPS_3$ crystals were investigated by means of X-ray diffraction (XRD, Cu-$K_\alpha$ radiation at 50 kV). The exfoliated flakes and devices were characterized by using an optical microscope (Nikon, Tokyo, Japan, ECLIPSE LV100ND) and scanning electron microscopy (SEM, Zeiss sigma300). A Bruker Dimension Icon atomic force microscopy (AFM, Karlsruhe, Germany) was utilized to test thicknesses and morphology. Raman spectra was acquired on an HR 800 Jobin Yvon Horiba by using a 532-nm laser excitation. The electrical and photoelectronic performances of the devices were measured by using a probe station (Cascade Microtech Inc., Beaverton, OR, USA, EPS150) for room temperature tests, a semiconductor analyzer (Agilent, Palo Alto, CA, USA, B1500A), a probe station

(Cascade M150), and a laser diode controller (Thorlabs, Newton, NJ, USA, ITC4001, with laser excitations of 405, 516, 638, 800.5, and 1065 nm) for low temperature tests.

## 3. Results

### 3.1. Characterization of Few-Layered NiPS$_3$ Flakes

The few-layered NiPS$_3$ crystal structure from the side view is shown in Figure 1a. Inside the layer, each bonded unit of the $[P_2S_6]^{4-}$ bipyramids is located in the center of the nearby honeycomb arrangement of divalent nickel (Ni) ions, and connected with the Ni atoms through six sulfur (S) atoms. The interlayers are connected via vdWs force. In general, the interlayer spacing is defined as the sum of one layer thickness and a large vdWs gap between the adjacent layer, which is estimated to be 6.35 Å [18]. Compared to graphite, NiPS$_3$ has a smaller calculated cleavage energy value, which indicates more feasibility to obtain few-layered flakes through the mechanical exfoliation method. From the XRD pattern in Figure 1b, the quality of bulk NiPS$_3$ crystals (inset, Figure 1b, shiny black, 3 mm × 3 mm) can be verified due to all the diffraction peaks being perfectly consistent with the reported data (PDF Card #33-0952) and show vdWs stacking along the *c* axis. According to the AFM image (Figure 1c), it exhibited the various thicknesses over a considerably large area and layered structures, and the thinnest thickness was 3.16 nm, corresponding to 5 layers.

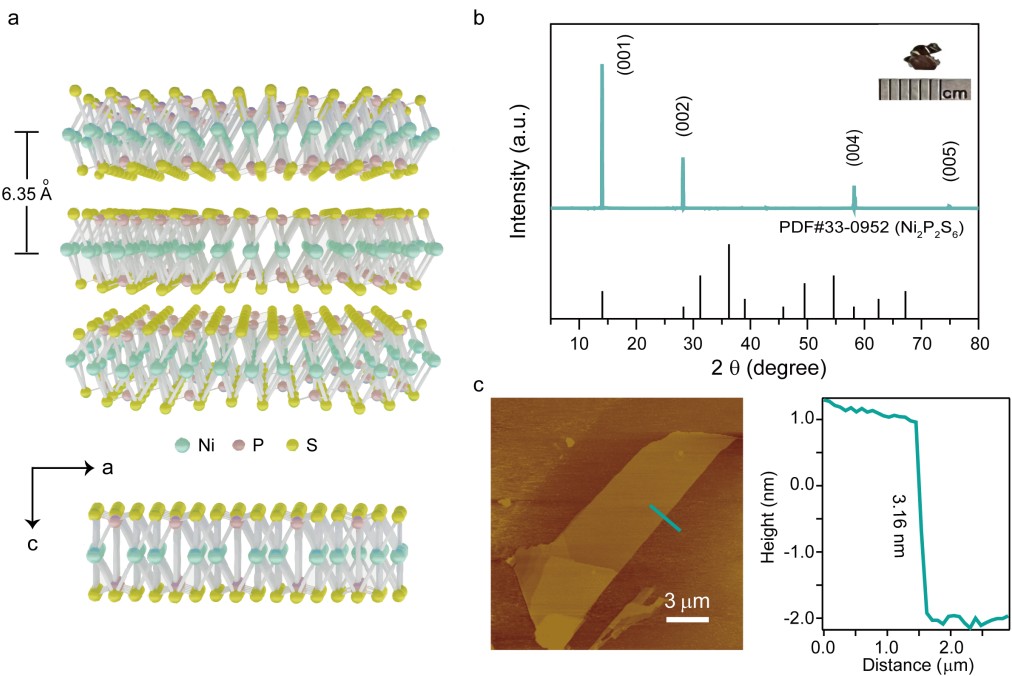

**Figure 1.** Structural characterizations of NiPS$_3$. (**a**) Schematics of the crystal structure of NiPS$_3$, and the blue, pink, and yellow spheres represent Ni, P, and S atoms, respectively. (**b**) The XRD pattern of the bulk NiPS$_3$ single-crystal along with the standard pattern. (**c**) Atomic force microscopic (AFM) topographic image of a typical few-layered NiPS$_3$ flakes, and the corresponding height profile measured across NiPS$_3$ flakes.

To further research the electrical properties and photoresponse performance, the field effect transistors (FET) of few-layered NiPS$_3$ are required to be fabricated. Considering the air stability and carrier mobility of NiPS$_3$ FET devices, hBN flakes were extensively selected to encapsulate sample and be as the substrate as well. A typical hBN-encapsulated NiPS$_3$ device contacted with Au electrodes is shown in Figure 2a. The AFM image along with the height profile are shown in Figure 2b,c, it is clear that hBN flakes fit tightly with few-layered NiPS$_3$ with the thickness of 10.4 nm, consistent with the most usually chosen thickness. In Figure 2d, Raman spectra of the BN-encapsulated NiPS$_3$ heterojunction measurements with

a 532-nm laser, including five in-plane $E_g$ modes and three out-of-plane $A_g$ modes, show that few-layered samples can be protected for a month as well as its pristine ones (orange line). All the electrical transport measurement were carried out in a vacuum chamber and can be repeated. As is given in Figure 2e, we plotted the current-voltage curves at different gate voltages, which exhibited non-linear behaviors attributed to a high contact resistance and the Schottky barrier formation at the interface between the $NiPS_3$ semiconductor and metal electrodes [23]. Moreover, the n-type transport behavior can be obviously verified from the transfer curves in Figure 2f, the source-drain current increased with the increasing positive gate voltage. In contrast with reported $NiPS_3$ FET, the electrical performances of hBN-encapsulated $NiPS_3$ FET have been greatly improved [29].

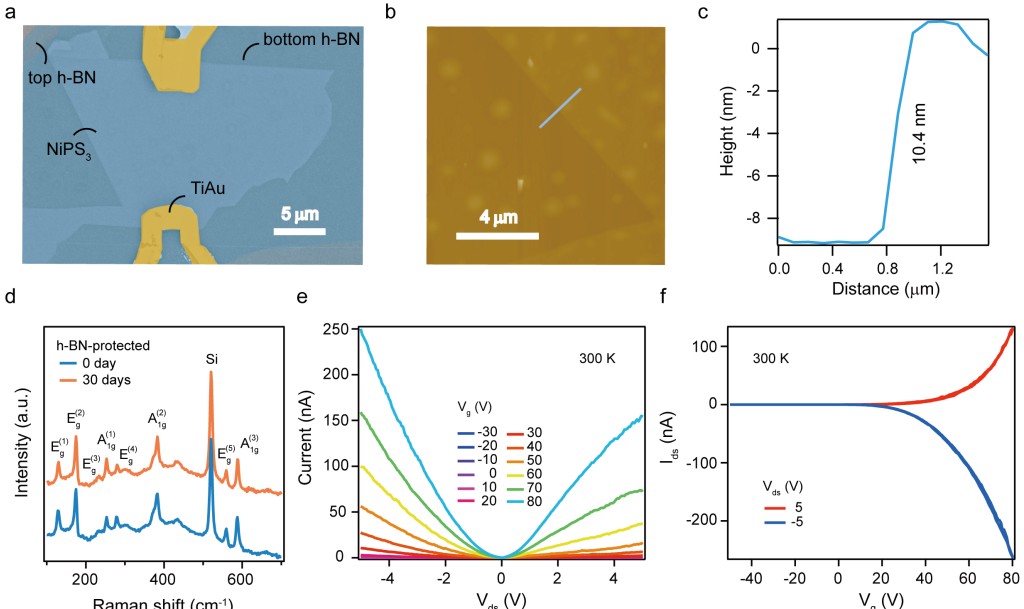

**Figure 2.** Electrical properties of the few-layered $NiPS_3$ field effect transistor. (**a**) False-colored scanning electron micrograph (SEM) image of the few-layered $NiPS_3$ device. (**b**) AFM image of vdWs heterojunction in (**a**). (**c**) The corresponding height profile of the heterojunction along the blue line in (**b**). (**d**) Raman spectra of the hBN-encapsulated $NiPS_3$ obtained after 0 day (blue solid line) and 30 days (orange). (**e**) The current-voltage characteristics at different gate voltages ($V_g$) as given in the figure for the $NiPS_3$ FET device at 300 K. $V_{ds}$ is the source-drain voltage. (**f**) The transfer curves at 5 V and −5 V source-drain voltage for the same sample in (**e**) at 300 K. $I_{ds}$ is the source-drain current.

## 3.2. Photoresponses at Different Temperatures

The demonstration of the $NiPS_3$ phototransistor is shown in Figure 3a. To evaluate the photoelectric performances of the $NiPS_3$ FET, the photoresponse measurements were systematically conducted under light illumination with various wavelengths of 405, 516, 638, and 800.5 nm, corresponding powers are 0.64, 2.60, 99, and 45 mW, respectively. As presented in Figure 3b,c, we first performed experiments at liquid nitrogen temperature of 77.3 K. Notably, the FET device had a detectable photoelectric response under intense radiation, but only an extremely weak photocurrent ($I_{ph}$, $I_{ph} = I_{light} - I_{dark}$) when laser excitation changed to 1065 nm, revealed a broad spectral response from near infrared to ultraviolet radiation. Additionally, this effect can be efficiently tuned by both the gate voltage ($V_g$) and source-drain bias ($V_{ds}$), the maximum photoresponse was achieved at 70 V gate voltage and −5 V bias voltage.

Considering differences in laser power (*P*), we calculated the photoresponsivity (*R*) through the following Equation (1) [30]:

$$R = \frac{I_{ph}}{PS},\tag{1}$$

where $I_{light}$ and $I_{dark}$ indicate the source-drain current under light illumination and in the dark state, *P* is the light power, *S* is the effective area of the device. In Figure 3d, the photoresponsivity dependence of laser wavelength behaves like a trivial phenomenon [31], *R* decreased with increasing wavelength from 405 nm to 800.5 nm, lasers above 800.5 nm (1.5 eV) were inadequate to excite the electrons' transition from the valence band to the conduction. Namely, the cut-off wavelength was beyond 800.5 nm, the reason can be ascribed to the existence of a large enough surface state acting as middle levels in the forbidden band [22,32,33]. Furthermore, the photoswitching behavior under illumination with different wavelengths had also been developed, and the determined rise time was illustrated in Figure 3d (blue), faster response for higher excitation energy. Figure 3e,f link with a 405-nm laser and displayed a minimum rise time of 30 ms.

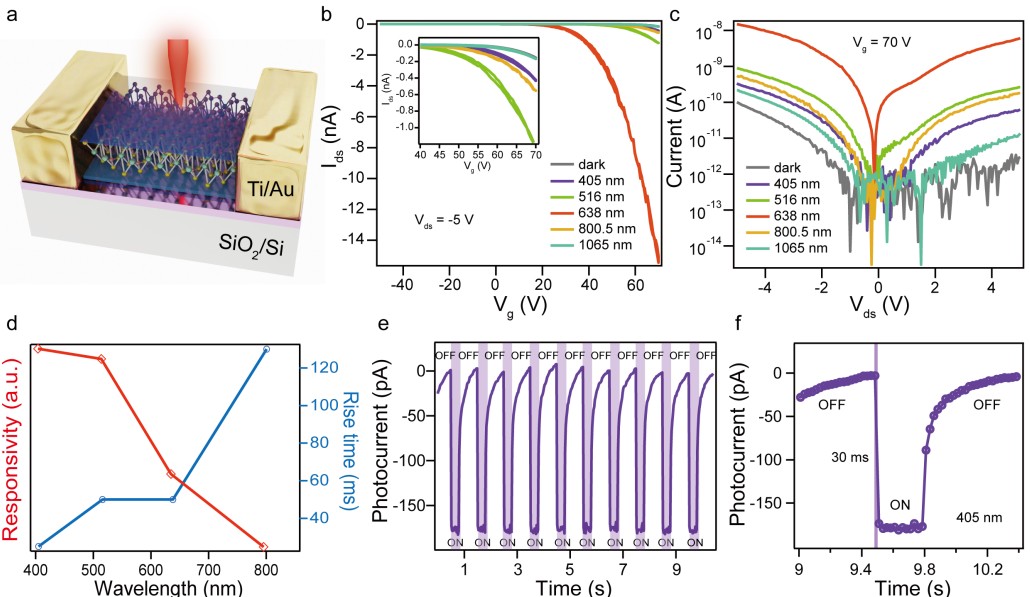

**Figure 3.** Photoelectric performances of the few-layered NiPS$_3$ photodetector at 77.3 K. (**a**) Demonstration of the NiPS$_3$ phototransistor. (**b**) The transfer curves at −5 V source-drain voltage in dark and under illumination. (**c**) Current-voltage curves at 70 V gate voltages in the dark and under illumination. (**d**) Photoresponsivity (red) and rise time (blue) as a function of wavelength at $V_{ds} = -5$ V and $V_g = 70$ V. (**e**) Photoswitching behavior under illumination with a wavelength of 405 nm. $V_{ds} = -5$ V and $V_g = 70$ V. (**f**) Photoresponse time of the NiPS$_3$ detector, and rise time is 30 ms.

In Figure 4a–c, the magnitude of photoresponse based on NiPS$_3$ FET varied with temperature. Then we extracted the values of $I_{light}/I_{dark}$ at both 70 V gate voltage and ±5 V bias voltage (Figure 4d). This results suggested a decreasing trend of the on-off ratios following the temperature, regardless of external electric field and wavelengths. In order to rationalize our experimental results, the carrier mobility (*μ*) and Schottky barrier height (SBH, $\varphi_b$) were calculated via the following equations [34,35]:

$$\mu = \left[\frac{dI_{ds}}{dV_g}\right] \times \left[\frac{L}{WC_iV_{ds}}\right], \tag{2}$$

$$\frac{dV}{d\ln I_{ds}} = n\frac{kT}{q} + I_{ds}R_S, \tag{3}$$

$$H(I_{ds}) = V_{ds} + n\frac{kT}{q}\ln\left(\frac{I_{ds}}{AA^*T^2}\right), \tag{4}$$

$$H(I_{ds}) = n\varphi_b + I_{ds}R_S, \tag{5}$$

where *L* is the conductive channel length, *W* is the conductive channel width, $C_i$ is the capacitance between NiPS$_3$ flakes and back gate per unit area, n is the ideality factor, *k* is the Boltzmann constant, *q* is the electronic charge, R$_S$ is the series resistance, *A* is the effective diode area, A* is the effective Richardson constant, given by A* = $4\pi q m_e k^2/h^3$ [36,37], *h* is the Planck constant, and m$_e$ is the effective mass of the majority carriers in the NiPS$_3$ (calculated by a first-principles method [38]), *T* is the temperature measured the current-voltage curves, and $\varphi_b$ is the zero-bias barrier height. Generally, we use the thermionic emission model (Cheung's functions (3)–(5)) based on the current-voltage curves at different temperatures. For few-layered NiPS$_3$ FET, the enhancement in carrier mobility happened with increasing temperature (Figure 4e), as did the Schottky barrier height; however, the decreasing trend in series resistance (Figure 4f). In addition, there was an agreement with the R$_S$ values determined by Cheung's two methods ((3) and (5)).

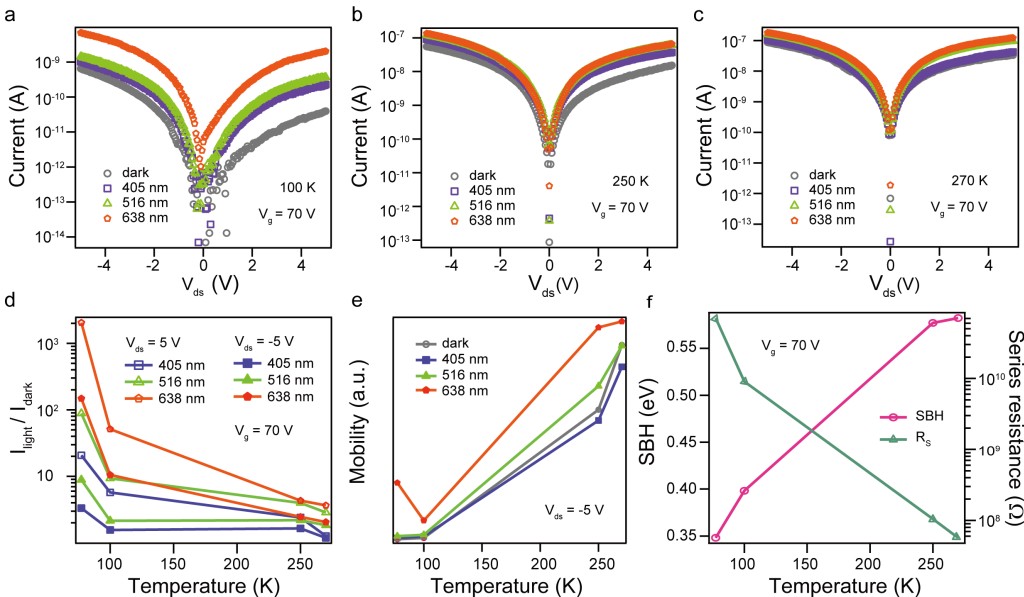

**Figure 4.** Temperature-dependent photoresponse of the NiPS$_3$ photodetector. (**a–c**) The current-voltage curves of the NiPS$_3$ device at 100, 250, and 270 K, respectively. (**d**) Photoresponsivity as a function of temperature under different light illumination (405, 516, and 638 nm). $V_{ds} = -5$ V and $V_g = 70$ V. (**e**) The carrier mobility for temperature ranging from 77.3 to 270 K, at external bias of $-5$ V. (**f**) Temperature dependent Schottky barrier height (SBH) (rose red) and series resistance (R$_S$) (green) at gate voltage of 70 V, in the dark.

## 4. Discussion

Obviously, the dark current serves as the baseline of photoresponse, when a phototransistor mainly used to sense the presence of the incident light, the maximum value of the I$_{light}$/I$_{dark}$ is chased. On the one hand, the photogenerated current relies on the absorption of photons in the NiPS$_3$ flakes and the quantities of electrons extracted from a single photon. The strong interactions with the incident light have been confirmed in the majorities of two-dimensional semiconductors despite of atomic thickness. Based on our findings, the lower Schottky barrier induced more photogenerated electrons to transport between the metal electrodes and the conductive channel at low temperature. On the other hand, the dark current grew by two orders of magnitude with the rise of temperature, caused by mobility improvement and resistance reduction. As a consequence, the on–off ratios were lower and lower as the temperature increased.

From a device architecture perspective, these behaviors of temperature dependencies can be understood by the existence of Schottky barrier inhomogeneities [39,40]. In addition, thermal fluctuations play a key role in the arising of a free carrier. Effectively, there is a competitive relation between Schottky barrier height and thermal fluctuations with chang-

ing temperature. In practice, the square of channel length in phototransistors is inversely proportional to quantum efficiency, and reducing the channel length is a straightforward method to achieve higher photosresponse [41]. Enhancing absorption of photons via quantum dots, and raising contact performances at the metal–semiconductor interface can yet be regarded as a convenient approach to explore given potential applications in flexible and ultrasensitive UV phototransistors over a broad temperature range.

## 5. Conclusions

In conclusion, we have demonstrated a typical hBN-encapsulated $NiPS_3$ phototransisitor via mechanical exfoliation and dry-transfer method. The protected $NiPS_3$ devices exhibited strong stability under ambient conditions over a month, and almost ideal electrical properties at room temperature. From a photoelectric point of view, the few-layered $NiPS_3$ displayed a broad photoresponse at liquid nitrogen temperature, and this performance depended on temperatures as well. Our results clearly demonstrate a significant contribution to the research expounding the role of the temperature in photodetectors.

**Author Contributions:** Conceptualization, methodology, formal analysis, writing–original draft preparation, X.S.; sample fabrication, data curation, discussion, Y.L. and X.S. All authors have read and agreed to the published version of the manuscript.

**Funding:** This research was funded by the National Natural Science Foundation of China (NSFC) (Nos. 11974357 and 12204490).

**Institutional Review Board Statement:** Not applicable.

**Informed Consent Statement:** Not applicable.

**Data Availability Statement:** Data is available from the corresponding author on reasonable request.

**Acknowledgments:** The authors appreciate the help of Maolin Chen in carrying out photoelectric performances measurements.

**Conflicts of Interest:** The authors declare no conflict of interest.

**Sample Availability:** Samples of the compounds $NiPS_3$ are available from the authors.

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
