# Peer review of "Phototransistors Based on hBN-Encapsulated NiPS3"

_magnetochemistry, doi:10.3390/magnetochemistry8090101_

Round 1

Reviewer 1 Report

1. The key findings of the intended work is to furnish the ultra sensitive light detection from few layered hBN-encapsulated NiPS3 field effect transistors. Therefore, the title of the manuscript supposed to be reframed, accordingly. Also, reframe the conclusions to highlight the potential outcomes of the work (and supposed to be in-line as narrated in the abstract).  Moreover, describe the key advantages of operating the said devices at the liquid nitrogen temperature. Why this temperature is so crucial?  It is suggested to briefly narrate the terms like Schottky barrier height/Schottky barrier at once where they have introduced at first (or cite suitable refs.).

2. There should be spacing between the text and the cited references. Apply full stop after the square bracket used for citing the references. Check the sentences (for example, see lines 9, 23, 52-53, 131, 143, 169, 174, 185) for spelling mistakes.   

3. Define hBN once in line 3.

4. Correct the chemical formula in line 64. 

5. In line 71 and 108, remove hyphen to have consistency in typing hBN (as earlier there was no hyphen).

6. Define sccm once (in line 72).

7. Define h as introduced in Richardson constant (see line 159). Earlier q is defined as electronic charge (in line 158) while in line 160 it is termed as Planck constant. Rectify the discrepancy in notation. 

8. Briefly narrate the mechanical exfoliation method and dry-transfer method (or cite suitable references). 

9. Define Vbg and Ids in context to Eq. (2) and (3), respectively. 

10. Define H(Ids) in Eq. (4) and (5) once for the sake of readers.

11. Be sure regarding the unit of mobility as defined in Fig. 4e along y-axis.

12. Authors have reported the thickness of 10 nm in abstract while in line 109 thickness of 10.4 nm is reported. Justify the choice of thickness.   

Reviewer 2 Report

In this manuscript, the authors presented the phototransistors based on the few-layered NiPS3. Overall, the experimental results were well organized and the related interpretations look reasonable. However, some comments should be explained. I recommend to publish this manuscript with major revision.

1.     In the method, the authors should provide the schematic diagram of the fabrication. This one makes the reader realize how to fabricate the few-layered NiPS3.

2.     Because the authors use 3M Scotch tape to exfoliate the NiPS3, how about the uniformity of the few-layered NiPS3?

3.     In addition, the authors utilize exfoliation to obtain the few-layered NiPS3. Usually, the few-layered NiPS3 has a lot of defects to degrade the performance of the phototransition. How do the authors ensure no defect in the few-layered NiPS3? If the defects exist in the few-layered NiPS3, please mention what kind the defects are and discuss how to affect the performance of the phototransition.

4.     The authors should provide the conversion efficiency of the phototransition.

Reviewer 3 Report

In this work, the authors experimentally explores the photoresponse of few layer hBN encapsulated NiPS3 transistors in low temperatures also in room temperature and over a period of a month. Because certain, actually most quantum applications require ultralow temperatures, this is clearly a significant contribution to the field especially considering practical applications. Both the fabrication and characterization methods are solid, and well explained in the manuscript. The manuscript is prepared well in general, though requiring some grammar and typo fixings, such as “resisitance" just before Section 2.

I think the physical reason of inducing more photogenerated electrons due to lower Schottky barrier needs a good explanation.

Also, to provide a better insight, some relevant works on hBN encapsulated NiPS3 at low temperature should be mentioned, such as the one on photoluminescence [ Nature Nanotechnology volume 16, pages 655–660 (2021)].

Round 2

Reviewer 2 Report

This paper can be accepted for this journal now.